# Luminescence Properties of Green Phosphor Ca_2_Ga_2_(Ge_1-x_Si_x_)O_7_:y%Eu^2+^ and Application

**DOI:** 10.3390/ma16103671

**Published:** 2023-05-11

**Authors:** Xiangqian Kong, Zhihua Qiu, Lina Wu, Yunfei Lei, Lisheng Chi

**Affiliations:** 1College of Chemistry and Materials Science, Fujian Normal University, Fuzhou 350116, China; xq20kong@fjirsm.ac.cn (X.K.); 107062019105@student.fjnu.edu.cn (Y.L.); 2Fujian Science and Technology Innovation Laboratory for Optoelectronic Information of China, Fuzhou 350116, China; nl20wu@fjirsm.ac.cn; 3Fujian Key Laboratory of Fuel and Materials in Clean Nuclear Energy System, Fujian Institute of Research on the Structure of Matter, Chinese Academy of Sciences, Fuzhou 350002, China; 4College of Chemistry, Fuzhou University, Fuzhou 350116, China

**Keywords:** phosphor, luminescence property, high-temperature solid-state reaction, Eu^2+^

## Abstract

Rare earth luminescent materials demonstrate significant advantages in lighting and energy saving, and detection etc. In this paper, a series of Ca_2_Ga_2_(Ge_1-x_Si_x_)O_7_:y%Eu^2+^ phosphors were synthesized by high-temperature solid-state reaction and characterized by X-ray diffraction and luminescence spectroscopy methods. The powder X-ray diffraction patterns reveal that all the phosphors are isostructural with a space group of *P*$\bar{4}$21*m*. The excitation spectra of Ca_2_Ga_2_(Ge_1-x_Si_x_)O_7_:1%Eu^2+^ phosphors exhibit significant overlapping of the host and the Eu^2+^ absorption bands, which facilitates Eu^2+^ absorbing the energy to increase its luminescence efficiency when excited by visible photons. The emission spectra show that the Eu^2+^ doped phosphors have a broad emission band with a peak centered at 510 nm arising from the 4f^6^5d^1^→4f^7^ transition. Variable temperature fluorescence reveals that the phosphor has a strong luminescence at low temperature but has a severe thermal quenching effect when temperature rises. The optimal Ca_2_Ga_2_(Ge_0.5_Si_0.5_)O_7_:1.0%Eu^2+^ phosphor shows promise for application in the field of fingerprint identification based on the experimental results.

## 1. Introduction

Rare earth luminescent materials have great advantages in energy saving, high efficiency and environmental protection, so they have been widely used in medical treatment, lighting, detection and so on [1,2,3,4]. Rare-earth-doped inorganic materials are usually synthesized by high-temperature solid-state reaction. Rare earth ions are introduced to occupy the lattice in traditional inorganic materials to emit fluorescence. Rare earth elements have excellent luminescence properties and energy conversion capability. Therefore, inorganic materials doped with small amounts of rare earth elements can emit abundant photons [5,6]. Particularly, the phosphor emitting green color has a special value in the field of biometrics. As Eu^2+^ ions are characteristic of the 5d-4f electronic structure, Eu^2+^ ions doped in inorganic materials show a strong absorption band and high luminous efficiency. When entering the inorganic material lattice under the reduction atmosphere, 5d-level electrons will undergo a 5d-4f transition, leading to Eu^2+^ emitting visible emission [7,8].

Jiao et al. [9] studied luminescence properties of different rare earth ions doped in Ca_2_Ga_2_SiO_7_. In the work, the fluorescence efficiency of the Eu^2+^ doped material is relatively low. Zou et al. [10] studied the effect of various Ge/Si ratios in Ca_2_Ga_2_(Ge,Si)O_7_ material on the fluorescence efficiency of the rare earth ions. However, the luminescent properties of Eu^2+^ in Ca_2_Ga_2_(Ge,Si)O_7_ have not been explored yet.

Here, we studied the effect of different Ge/Si ratios on the luminescence property of Eu^2+^ in the Ca_2_Ga_2_(Ge,Si)O_7_ material with an aim to obtain a fluorescent material with high luminous efficiency. The experimental results show that the obtained phosphor has decent luminescence properties, with promise for application in fingerprint identification at room temperature.

## 2. Materials and Methods

In this study, CaCO_3_ (99.99%, Aladdin, Shanghai, China), Ga_2_O_3_ (99.99%, aladdin), GeO_2_ (99.99%, aladdin), SiO_2_ (99.99%, aladdin) and Eu_2_O_3_ (99.9%, aladdin) were used as starting materials. A series of Ca_2_Ga_2_(Ge_1-x_Si_x_)O_7_:y%Eu^2+^ (x = 0–1, y = 0–2) was synthesized by conventional high-temperature solid-state reaction in a tube chamber furnace BTF-1400C (BEQ, Anhui, China) under 20%H_2_-80%Ar atmosphere. CaCO_3_, Ga_2_O_3_, GeO_2_, SiO_2_ and Eu_2_O_3_ were stoichiometrically weighed using an analytical balance PWN85ZH (Ohaus, Shanghai, China) and mixed and grinded in an agate with 15 mL ethanol (99.7%, Sinopharm, Beijing, China) until ethanol evaporated. Then, the mixture was loaded into a corundum boat that was placed in the high-temperature tube furnace, followed by sintering in the reducing atmosphere at a temperature between 1100 °C and 1350 °C for 6 h. The used temperature increases with increase in Si content. After the furnace was cooled down to room temperature, the material was ground and used for various characterizations.

The obtained materials were characterized by X-ray diffraction (XRD) method to determine their phase identity. The XRD data were measured on a Rigaku Mini-Flex II powder diffractometer (Rigaku, Tokyo, Japan)(Cu-K_α_, radiation at *λ* = 1.5406 Å) in the 2θ range of 10–80°. The excitation and emission luminescence spectra, quantum efficiency and PL decay curves were measured using FLS1000 fluorescence spectrophotometer (Edinburgh Instruments Ltd., Edinburgh, UK). Morphology and energy-dispersive X-ray spectra of materials were measured on a field emission scanning electron microscope (FESEM) SU-8010 (Hitachi, Tokyo, Japan)/EDX (Oxford Instrument, Abingdon, UK).

## 3. Results and Discussion

### 3.1. Structure Analysis of Ca_2_Ga_2_SiO_7_ and Ca_2_Ga_2_GeO_7_

The crystallographic data for Ca_2_Ga_2_SiO_7_ [9,11] and Ca_2_Ga_2_GeO_7_ [6] are listed in Table 1. It is shown that both compounds belong to a tetragonal system with a space group of *P*$\bar{4}$2_1_*m*, so they are isostructural. The structure of Ca_2_Ga_2_GeO_7_ is plotted in Figure 1a. It contains two types of tetrahedral coordination: one tetrahedral site is occupied by Ga, while another one is equally occupied by Ge and Ga. These GaO_4_ and (Ge, Ga)O_4_ tetrahedra share corners with each other to form two-dimensional layers in the ab plane, which stack along the c axis to form a three-dimensional structure, with Ca^2+^ occupying the interstitials between layers [12,13]. Figure 1b shows the XRD patterns of Ca_2_Ga_2_(Ge_1-x_Si_x_)O_7_:y%Eu^2+^ (x = 0, 0.3, 0.5, 0.7, 1; y = 1). All the XRD patterns of Ca_2_Ga_2_(Ge_1-x_Si_x_)O_7_:y%Eu^2+^ (x = 0–1; y = 0–2) are provided in Appendix A. The figures show that all the peaks in these XRD patterns match well with those in the standard PDFs, 74-1608 for Ca_2_Ga_2_SiO_7_ or 38-1328 for Ca_2_Ga_2_GeO_7_, without exception [6]. Therefore, the samples with a composition of Ca_2_Ga_2_(Ge_1-x_Si_x_)O_7_:y%Eu^2+^ prepared under the reduction conditions in this study all were determined to be single-phase.

Because the radius of Si^4+^ (0.26 Å) is smaller than that of Ge^4+^ (0.39 Å) [14], when x increases from 0 to 1, all the peaks shift to a higher angle according to Bragg’s law. For example, in Figure 1b, the strongest peak is at 30.591° when x = 0 shifts toward 30.980° at x = 1. As a result, the cell volume for Ca_2_Ga_2_SiO_7_ is smaller than that of Ca_2_Ga_2_GeO_7_, as shown in Table 1. Investigation into the XRD patterns reveals that doping of Eu^2+^ with up to 2% of Ca_2_Ga_2_Ge_0.5_Si_0.5_O_7_ did not change its structure. As the radius of Eu^2+^ (r = 1.20 Å) is similar to Ca^2+^ (r = 1.06 Å) [7], Eu^2+^ is doped into Ca_2_Ga_2_Ge_0.5_Si_0.5_O_7_ to replace the position of Ca^2+^ to maintain the charge balance of the compound and leave the structure unchanged.

Figure 1c shows the EDX of Ca_2_Ga_2_(Ge_0.5_Si_0.5_)O_7_:1%Eu^2+^. This indicates that Eu^2+^ was successfully doped into Ca_2_Ga_2_Ge_0.5_Si_0.5_O_7_. From Figure 1b,c, the conclusion is reached that Eu^2+^ ions have successfully replaced Ca^2+^ in Ca_2_Ga_2_(Ge_0.5_Si_0.5_)O_7_:1%Eu^2+^ without changing its structure. The morphological features of Ca_2_Ga_2_(Ge_0.5_Si_0.5_)O_7_:1%Eu^2+^ are shown in Figure 1d and Appendix A. The image shows that the particles are irregular and the dimension is in the range from 1 µm to 10 µm. However, currently, there is no strict requirement regarding particle size of the phosphors for an application [15].

### 3.2. Spectroscopic Study of Ca_2_Ga_2_(Ge_1-x_Si_x_)O_7_:1%Eu^2+^

The excitation and emission spectra of Ca_2_Ga_2_(Ge_1-x_Si_x_)O_7_:y%Eu^2+^ were measured using the FLS1000 fluorescence spectrophotometer. The emission spectra of Ca_2_Ga_2_(Ge_1-x_Si_x_)O_7_:1.0%Eu^2+^ (x = 0.3, 0.5, 0.7, 1) are shown in Figure 2a. All the emission spectra and the excitation spectra of Ca_2_Ga_2_(Ge_1-x_Si_x_)O_7_:1.0%Eu^2+^ (x = 0–1) are presented in Appendix A. It is worth noting that both excitation spectra and emission spectra are measured under the same conditions. The excitation spectra were collected in the wavelength range from 250 nm to 450 nm under the 510 nm emission, while the emission spectra were collected in the wavelength range from 400 nm to 800 nm under the 333 nm excitation. Figure 2a shows that when x is at 0.3, there is a very weak peak observed at 510 nm, as shown in the inset. Therefore, the measurement of the excitation spectrum started with the material composition at x = 0.4. All the excitation spectra are characteristic of two broad bands, with one located at 273 nm and another one located at 333 nm. These two observed excitation peaks are attributed to the absorption of Ca_2_Ga_2_(Ge_1-x_Si_x_)O_7_:1.0%Eu^2+^ host and the electron transition from the 4f^7^ ground state (^8^S_7/2_) to the 4f^6^5d^1^ excited state (T_2g_) of doped Eu^2+^, respectively [16]. Appendix A shows significant overlapping of the two broad excitation bands, which facilitates Eu^2+^ absorbing the energy to increase luminescence efficiency when excited by visible photons. Figure 2a shows that all the emission spectra have a broad band ranging from 430 nm to 730 nm with the highest peak located at 510 nm. The green emission is ascribed to the 4f^6^5d^1^→4f^7^ transition of Eu^2+^ doping in the Ca^2+^ site [17], which is highly influenced by the crystal field generated due to the electron–electron interaction between the coordination oxygen atoms and the luminous centers. In addition, there are no emission peaks of Eu^3+^ observed in any of the samples [18,19]. This indicates that Eu^3+^ ions in the starting material Eu_2_O_3_ were completely reduced to Eu^2+^ by the high-temperature solid-state reaction in the 20%H_2_-80%Ar atmosphere.

Analyses on the emission spectra of Ca_2_Ga_2_(Ge_1-x_Si_x_)O_7_:1.0%Eu^2+^ reveal that the emission peak intensity is highly dependent on Si content. When x ≤ 0.1, there is no emission peak observed. The strongest emission was observed at x = 0.5, as shown in Figure 2a. In the same manner, fluorescence quantum efficiency can be used to express Si content dependence of the emission intensity, as it is defined as the number of photons emitted over the number of photons absorbed. Figure 2b shows that the fluorescence quantum efficiency increases with increasing x with a maxima value at x = 0.5, followed by decreasing and upturning at x = 0.6. The fluorescence intensity observed in Ca_2_Ga_2_(Ge_0.5_Si_0.5_)O_7_:1.0%Eu^2+^ is more than 3 orders higher than that at x ≤ 0.3. This can be ascribed to the change in band gap between the valence band and conduction band in the Ca_2_Ga_2_(Ge_1-x_Si_x_)O_7_ host when Ge^4+^ is substituted by Si^4+^.

Zou et al. [10] revealed that the valence band and conduction band in Ca_2_Ga_2_SiO_7_ are dominated by the Ga, Si and O components, respectively. As the electron affinity of Si^4+^ is greater than that of Ge^4+^ [20], increasing the Si content in Ca_2_Ga_2_(Ge_1-x_Si_x_)O_7_:1.0%Eu^2+^ lowers the valence band, leading to the increase in the band gap. This is in good agreement with the adsorption spectra data by which the band gap was obtained showing a gradual increase from 4.81 eV to 5.05 eV with increasing x from 0 to 1.

Decrease in quantum efficiency of the materials with Si content increasing from 50% to 60% could be attributed to formation of the minor phase CaSiO_3_ [21], as the XRD results in Appendix A show enhancement in the intensity of the peaks at 24.06° and 29.02°.

In this section, the optimal luminescent material composition with Ca_2_Ga_2_(Ge_0.5_Si_0.5_)O_7_ was obtained by changing the ratio of Ge/Si while maintaining the doped Eu^2+^ concentration. In the following, the effect of Eu^2+^ concentration on the luminescent properties of the material with the ratio of Ge/Si at 0.5:0.5 will be studied.

### 3.3. Spectroscopic Study of Ca_2_Ga_2_Ge_0.5_Si_0.5_O_7_:y%Eu^2+^

In order to determine the optimal concentration of Eu^2+^ doped in the material, a series of samples of Ca_2_Ga_2_(Ge_0.5_Si_0.5_)O_7_:y%Eu^2+^ (y = 0, 0.5, 1, 1.5, 2) were prepared in the reducing atmosphere. The excitation and emission spectra of the materials with different Eu^2+^ contents were measured and are presented in Figure 3a,b, respectively.

The emission spectra show that the intensity of the emission peak at 510 nm increases with increasing y, with a maximum at y = 1 because of high adsorption efficiency, followed by decreasing, as shown in Figure 3c, due to the concentration quenching [21]. It has been known that the emission intensity is dependent on the critical distance between the Eu^2+^ ions [10], which can be calculated using Equation (1).
(1)$$R_{c}=2{\left(\frac{3V}{4{\pi X}_{c}N}\right)}^{\frac{1}{3}}$$
where V is the unit cell volume, X_c_ is the concentration of Eu^2+^ generating the strongest luminescence intensity (1%) and N is the number of atoms per unit cell, which is 12.

Based on the parameters in Table 1 and the luminescence data obtained in this study, the critical distance for Ca_2_Ga_2_(Ge_0.5_Si_0.5_)O_7_:1%Eu^2+^ is calculated to be 17.175 Å. As both Ca_2_Ga_2_GeO_7_ and Ca_2_Ge_2_SiO_7_ are isostructural, the average value of 318.156 Å^3^ for both cell volumes was used in the calculation. As the critical distance is much longer than 5 Å, the electric multipole–multipole interaction rather than exchange interaction predominates the energy transfer between the Eu^2+^ ions [16]. When the doped Eu^2+^ concentration is greater than 1%, the distance between the luminescence centers is less than the critical distance of 17.175 Å. This leads to a nonradiative energy transfer occurring between the luminescence centers, thus decreasing emission intensity.

Figure 3d shows the variations in the luminescence intensity of the Ca_2_Ga_2_(Ge_0.5_Si_0.5_)O7:y%Eu^2+^ samples with time. One can see that the luminescence decay curves can be well fit to the exponential model. Therefore, the luminescence lifetime can be calculated using Equations (2) and (3) [22].
(2)$$I\left(t\right)=A_{1}e^{\frac{-t}{\tau_{1}}}+A_{2}e^{\frac{-t}{\tau_{2}}}$$
(3)$$\tau =\frac{A_{1}\tau_{1}^{2}+A_{2}\tau_{2}^{2}}{A_{1}\tau_{1}+A_{2}\tau_{2}}$$
where A_1_ and A_2_ are the scalar quantity obtained from the curve fitting, I(t) represents the luminescence intensity at time t in ns, and τ_1_ and τ_2_ stand for lifetimes for different exponential components.

The luminescence lifetimes of Ca_2_Ga_2_(Ge_0.5_Si_0.5_)O_7_:y%Eu^2+^ were determined by fitting to 670 ns, 696 ns, 656 ns and 619 ns when y is at 0.5, 1, 1.5 and 2, respectively. All the luminescence lifetimes are in the range of 0.2–2 µs, which is the typical lifetime of Eu^2+^ emission.

Thermal stability of a phosphor must be evaluated for its application. Figure 3e shows the luminescence spectra of Ca_2_Ga_2_(Ge_0.5_Si_0.5_)O_7_:1.0%Eu^2+^ at different temperatures. The intensity of the emission peak at 510 nm gradually decreases with increasing temperature from 100 K to 600 K, as shown in Appendix A. It is shown that the thermal quenching of the phosphor is very significant, as the luminescence intensity at 600 K is reduced by 98% compared to that at 100 K. This effect can be ascribed to the electron–photon interaction under thermal disturbance, as the increased temperature facilitates the nonradiative transition of electrons at the 4f^6^5d^1^ excited state to the 4f^7^ ground state, leading to the decrease in the emission intensity. To further explore the thermal quenching mechanism, the activation energy is calculated using Equation (4) [23].
(4)$$\mathrm{In}\frac{I}{I_{0}}=\mathrm{InA}-\frac{\mathrm{Ea}}{\mathrm{kT}}$$
where *I* and *I*_0_ are the luminescence intensities at given temperature and normal temperature; A is the constant; and k is the Boltzmann constant (8.617 × 10^−5^ eV/K).

Based on the data obtained in Figure 3e, ln(*I*/*I*_0_) vs. 1/kT is plotted and shown in Appendix A. The activation energy of Ca_2_Ga_2_(Ge_0.5_Si_0.5_)O_7_:1.0%Eu^2+^ is calculated to be 0.247 eV. This suggests that the luminescence intensity of Ca_2_Ga_2_(Ge_0.5_Si_0.5_)O_7_:1%Eu^2+^ is greatly affected by temperature. When temperature rises, the electrons in the excited state can escalate to the higher vibration levels, from which radiative transitions cause emission band shifting to shorter wavelength, as observed in Figure 3e.

The photoluminescence (PL) stability of Ca_2_Ga_2_(Ge_0.5_Si_0.5_)O_7_:1.0%Eu^2+^ was tested under 333 nm irradiation at different times of 0, 5, 10, 20, 30, 45, 60, 75, 90, 105, 120, 135, 150, 165, 180, 195, 210 and 225 min [24,25]. The results presented in Figure 3f show that the fluorescence intensity of the Ca_2_Ga_2_(Ge_0.5_Si_0.5_)O_7_:1.0%Eu^2+^ phosphor decreased by 3% in the first 5 min, followed by maintaining invariance with ±1% fluctuations arising from the xenon lamp of the spectrometer. Therefore, Ca_2_Ga_2_(Ge_0.5_Si_0.5_)O_7_:1.0%Eu^2+^ phosphor has high PL stability.

Based on the experimental results obtained in this study, it is concluded that Ca_2_Ga_2_(Ge_0.5_Si_0.5_)O_7_:1.0%Eu^2+^ is the optimal phosphor. In the following section, application of the material in fingerprint identification is explored.

### 3.4. Ca_2_Ga_2_(Ge_0.5_Si_0.5_)O_7_:1.0%Eu^2+^ CIE Color and Application in Fingerprint Identification

The CIE coordinates of Ca_2_Ga_2_(Ge_0.5_Si_0.5_)O_7_:1.0%Eu^2+^ were calculated using the CIE 1931 software by importing the emission spectrum data measured under the excitation of 333 nm. Figure 4a presents the calculated coordinates in the CIE diagram. It is shown that the coordinates on the x and y axes were determined to be 0.257 and 0.515, respectively. It is concluded that Ca_2_Ga_2_(Ge_0.5_Si_0.5_)O_7_:1.0%Eu^2+^ is a green-emitting phosphor.

In nature, a living thing that comes into contact with solid matter leaves a trace such as a fingerprint [26]. When a fingerprint touches a glass surface, the phosphor will spread on the glass, and one can see the fingerprint under 365 nm violet light. Figure 4b shows an application of the Ca_2_Ga_2_(Ge_0.5_Si_0.5_)O_7_:1.0%Eu^2+^ material in fingerprint identification. You can clearly see a complete fingerprint and its five fine patterns, which are core point, ridge ending, bifurcation, island and pore. Different types of materials used for fingerprint identification are compared in Table 2 in terms of advantages and disadvantages. In addition, as the green phosphors are more widely used in fingerprint identification than other colors, Ca_2_Ga_2_(Ge_0.5_Si_0.5_)O_7_:1.0%Eu^2+^ exhibits its promise for application in fingerprint identification [27].

## 4. Conclusions

In conclusion, a green phosphor with the composition of Ca_2_Ga_2_(Ge_1-x_Si_x_)O_7_:y%Eu^2+^ was synthesized by high-temperature solid-state reaction in a reducing atmosphere and characterized by X-ray diffraction and luminescence spectroscopy methods. The highest emission intensity in Ca_2_Ga_2_(Ge_1-x_Si_x_)O_7_:y%Eu^2+^ is at the Ge/Si ratio of 0.5:0.5 with the doping concentration at 1%. The energy conversion between Eu^2+^ ions in the Ca_2_Ga_2_(Ge_0.5_Si_0.5_)O_7_:y%Eu^2+^ phosphor proceeds with an electric multipole–multipole interaction mechanism. The variable temperature fluorescence reveals that the fluorescence intensity of the material at low temperature is higher than that at the high temperatures. The Ca_2_Ga_2_(Ge_0.5_Si_0.5_)O_7_:1.0%Eu^2+^ material shows promise for application in fingerprint identification at room temperature.

## Figures and Tables

**Figure 1 materials-16-03671-f001:**
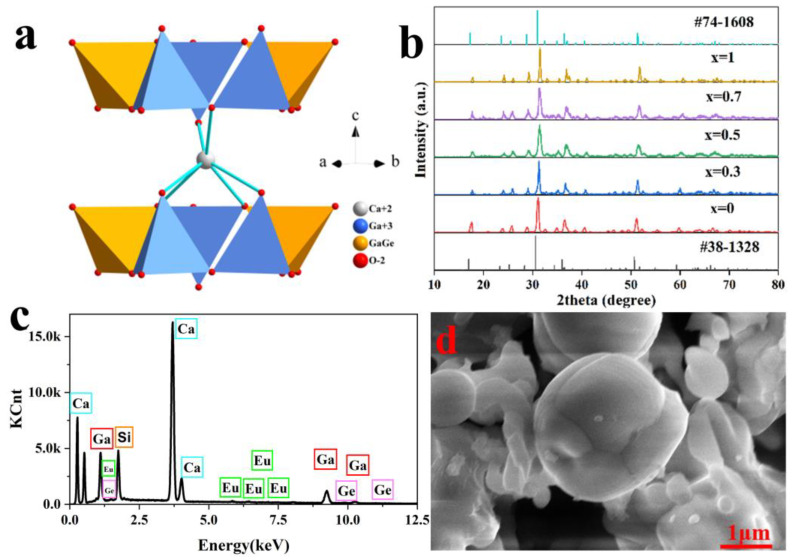
(**a**) The structure of Ca_2_Ga_2_GeO_7_. (**b**) XRD patterns of Ca_2_Ga_2_(Ge_1-x_Si_x_)O_7_:1%Eu^2+^. (**c**) EDX for Ca_2_Ga_2_(Ge_0.5_Si_0.5_)O_7_:1%Eu^2+^. (**d**) SEM image of Ca_2_Ga_2_(Ge_0.5_Si_0.5_)O_7_:1%Eu^2+^.

**Figure 2 materials-16-03671-f002:**
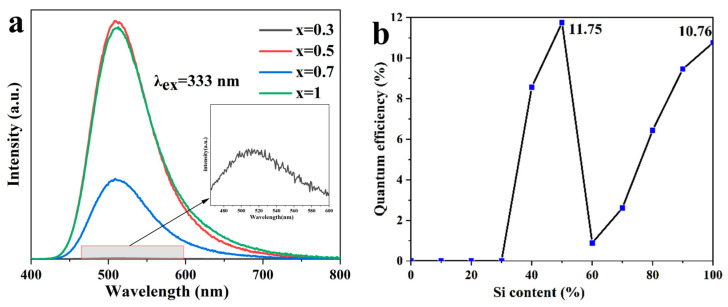
(**a**) Emission spectra of Ca_2_Ga_2_(Ge_1-x_Si_x_)O_7_:1.0%Eu^2+^. (**b**) Quantum efficiency vs. Si content.

**Figure 3 materials-16-03671-f003:**
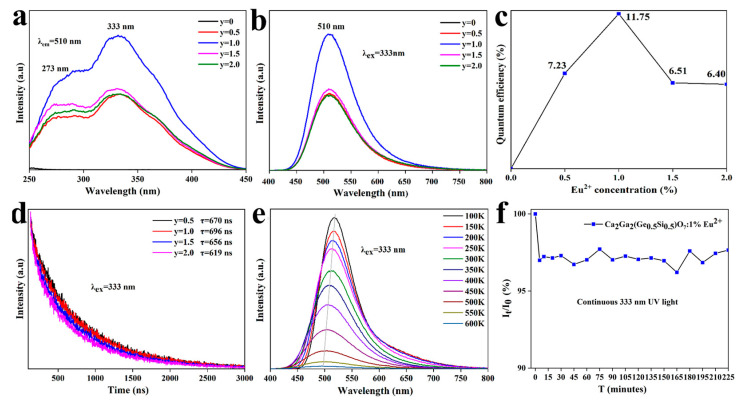
(**a**) Excitation spectra of Ca_2_Ga_2_(Ge_0.5_Si_0.5_)O_7_:y%Eu^2+^. (**b**) Emission spectra of Ca_2_Ga_2_(Ge_0.5_Si_0.5_)O_7_:y%Eu^2+^. (**c**) The relationship between quantum efficiency and concentration of Eu^2+^. (**d**) Emission intensity decay curves of Ca_2_Ga_2_(Ge_0.5_Si_0.5_)O_7_:y%Eu^2+^. (**e**) Variable temperature fluorescence emission spectra of Ca_2_Ga_2_(Ge_0.5_Si_0.5_)O_7_:1%Eu^2+^. (**f**) PL stability of Ca_2_Ga_2_(Ge_0.5_Si_0.5_)O_7_:1%Eu^2+^ under continuous 333 nm UV light.

**Figure 4 materials-16-03671-f004:**
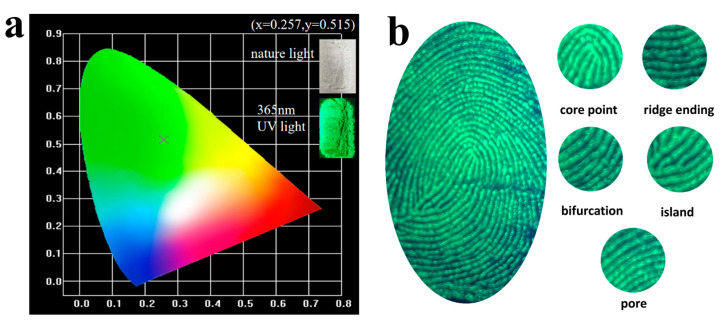
(**a**) CIE color coordinate for Ca_2_Ga_2_(Ge_0.5_Si_0.5_)O_7_:1.0%Eu^2+^. (**b**) Image of a fingerprint on glass under 365 nm UV light.

**Table 1 materials-16-03671-t001:** The crystallographic data for Ca_2_Ga_2_SiO_7_ and Ca_2_Ga_2_GeO_7_.

	Ca_2_Ga_2_SiO_7_ [9,11]	Ca_2_Ga_2_GeO_7_ [6]
PDF card number	74-1608	38-1328
System	tetragonal	tetragonal
Space group	*P*$\bar{4}$2_1_*m*	*P*$\bar{4}$2_1_*m*
Lattice parameters	a = b = 7.793 Å c = 5.132 Å	a = b = 7.896 Å c = 5.207 Å
Volume	311.671 Å^3^	324.640 Å^3^

**Table 2 materials-16-03671-t002:** Summary of materials applied in fingerprint identification [27].

Type of Material	Advantages	Disadvantages
Metal	High sensitivity, high conductivity	Low resistance to acids and bases
Metal oxide	Fingerprints with blood or sweat display well, high sensitivity	Low resistance to acids and bases
Quantum dots materials	Good photostability, old fingerprints display well	Expensive
Rare earth fluorescent materials	Wide range of colors, low background influence	Require additional illumination
This work	High sensitivity, high acid resistance, high PL stability	Require additional illumination

## Data Availability

The data presented in this study are available on request from the corresponding author.

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
