# Peer review of "Luminescence Properties of Green Phosphor Ca2Ga2(Ge1-xSix)O7:y%Eu2+ and Application"

_materials, 2023, doi:10.3390/ma16103671_

Round 1
Reviewer 1 Report
Reading the manuscript I did not find any particular issue.
The data are well presented and the research covers a topic of interest for the readership of the journal.
The labels of the figures are sometimes small it would be better to enlarge them.
The paper can be published.
-
Author Response
Dear Dr. Reviewer,
Thank you for reviewing materials-2361482.
Please find attached response to reviewer1 to see if you are satisfied with the response.
Thank you again for your time and comments.
Best Regards,
Lisheng Chi
Professor, Fujian Institute of Research on the Structure of Matter, CAS., China

Reviewer 2 Report
The section “Structure analysis of Ca2Ga2SiO7 and Ca2Ga2GeO7” should be revised. The authors give in it known literature data, namely information from the cards, images of fragments of the structure of the compound without giving references to structural works. The authors themselves have not determine the structure. The authors' own experimental data on the refinement of unit cell parameters are not availablens. ,
The calculation of the critical distance (to the third digit) also requires clarification. The value of the volume for the solid solution is unknown.
Author Response
Dear Dr. Reviewer,
Thank you for reviewing materials-2361482.
Please find attached response to reviewer2 to see if you are satisfied with the response.
Thank you again for your time and comments.
Best Regards,
Lisheng Chi
Professor, Fujian Institute of Research on the Structure of Matter, CAS., China

Reviewer 3 Report
This paper describes the synthesis and characterization of a series of Ca2Ga2(Ge1-xSix)O7:y%Eu2+ phosphors for potential use in lighting, energy-saving, and detection devices. The X-ray diffraction patterns indicate that all the phosphors have an isostructural arrangement with a space group of 15 P 42 1m. The Eu2+ absorption bands overlap with the host bands in the excitation spectra, facilitating efficient energy transfer to increase luminescence efficiency upon excitation by visible photons. The Eu2+ doped phosphors exhibit a broad emission band with a peak at 510 nm arising from the 4f65d1→4f7 transition. The variable temperature fluorescence reveals severe thermal quenching effects at high temperatures, while the phosphor has strong luminescence at low temperatures. The optimal Ca2Ga2(Ge0.5Si0.5)O7:1.0%Eu2+ phosphor shows promising application in fingerprint identification, based on the experimental results. This work provided interesting results and eventually can accept after minor corrections.
1. Comparing to others, what advantages of the prepared materials in synthesis and applications in this manuscript? A summary table is suggested.
2. Fig 2b, quantum yield varies by different Si content. Please discuss it in detail, such as why QY decrease in the Si content of 50% to 60%, however, increasing from 60 to 100%?
3. How about PL stability under light irradiation?
Author Response
Dear Dr. Reviewer,
Thank you for reviewing materials-2361482.
Please find attached response to reviewer3 to see if you are satisfied with the response.
Thank you again for your time and comments.
Best Regards,
Lisheng Chi
Professor, Fujian Institute of Research on the Structure of Matter, CAS., China

Round 2
Reviewer 2 Report
I recommend that the authors of the article provide in Table 1 their own data on the lattice parameters values of the compounds under study